# Generating and Evaluating Tests for K-12 Students with Language Model Simulations: A Case Study on Sentence Reading Efficiency

**Eric Zelikman\***
Stanford University
ezelikman@cs.stanford.edu

**Wanjing Anya Ma\***
Stanford University
wanjingm@stanford.edu

**Jasmine E. Tran**
Stanford University
jasetran@stanford.edu

**Diyi Yang**
Stanford University
diyiy@cs.stanford.edu

**Jason D. Yeatman**
Stanford University
jyeatman@stanford.edu

**Nick Haber**
Stanford University
nhaber@stanford.edu

## Abstract

Developing an educational test can be expensive and time-consuming, as each item must be written by experts and then evaluated by collecting hundreds of student responses. Moreover, many tests require multiple distinct sets of questions administered throughout the school year to closely monitor students' progress, known as parallel tests. In this study, we focus on tests of silent sentence reading efficiency, used to assess students' reading ability over time. To generate high-quality parallel tests, we propose to fine-tune large language models (LLMs) to simulate how previous students would have responded to unseen items. With these simulated responses, we can estimate each item's difficulty and ambiguity. We first use GPT-4 to generate new test items following a list of expert-developed rules and then apply a fine-tuned LLM to filter the items based on criteria from psychological measurements. We also propose an optimal-transport-inspired technique for generating parallel tests and show the generated tests closely correspond to the original test's difficulty and reliability based on crowdworker responses. Our evaluation of a generated test with 234 students from grades 2 to 8 produces test scores highly correlated (r=0.93) to those of a standard test form written by human experts and evaluated across thousands of K-12 students.

## 1 Introduction

Developing an educational test can be resource-intensive and time-consuming, as each item must be written by experts and then evaluated by collecting hundreds of student responses. This process of evaluating items in terms of properties like difficulty and their ability to discriminate between student abilities, known in psychometrics as **item calibration**, is fundamental to test development.

Furthermore, schools often require the creation of multiple, distinct test forms (a collection of

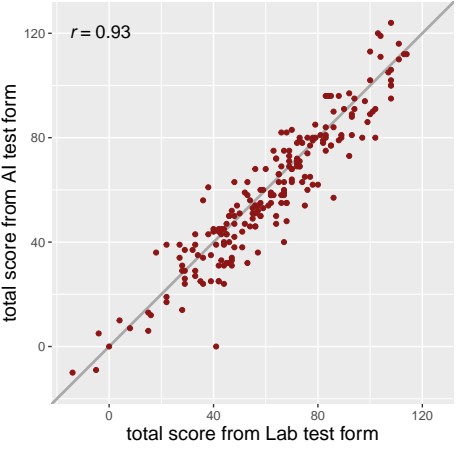

Figure 1: **Student test scores for human-generated vs model-generated tests.** Comparison of student scores on a test form of item-response-simulator-filtered language-model-generated items and a test form of human-generated items for a sentence reading efficiency test. The human-generated test form was designed by experts and calibrated across thousands of K-12 students; the AI test form was generated by GPT-4 and filtered by our proposed item-response simulator.

unique items) administered throughout the academic year, allowing for close monitoring of student progress while minimizing practice effects. These test forms, designed to be content-equivalent and reliably produce identical individual scores as the original test form, are known as **parallel tests**. Each step of the test development process, from expert item writing to extensive item calibration through large-scale data collection and ultimately parallel test creation and validation, poses significant demands in terms of resources and time. These challenges emphasize the necessity for an automated and efficient test development framework.

In response to the challenge of item development, many works have proposed leveraging language models to generate items for educational assessments and instruction (Agarwal et al., 2011; Stasaski et al., 2021; Srivastava and Goodman, 2021; Heck and Meurers, 2022; Rathod et al., 2022;

---

*These authors contributed equally to this work

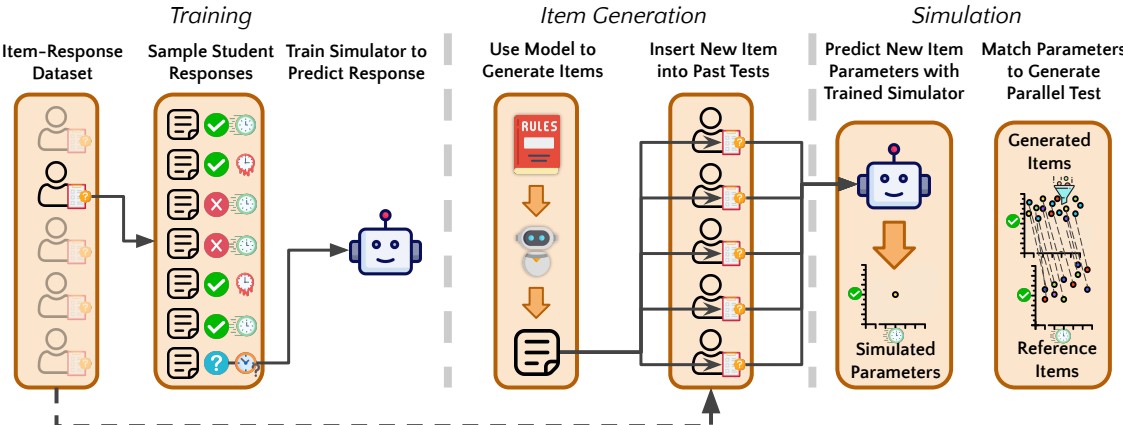

Figure 2: **Method overview.** We first train the simulator to predict a student's response to a question conditioned on their previous responses. Then, we generate new items and insert them into past tests for simulation. Finally, we perform simulation and match the new items to an existing test. Note the simulation and generation models differ.

Zou et al., 2022; White et al., 2022). However, estimating the relevance, quality, and difficulty of these generated items and test forms is an open challenge that must be carefully addressed for both psychometric and NLP communities as well as stakeholders in our education system. We propose an item-response simulator, training LLMs with student response data and simulating the responses of past participants on new items in order to calibrate them.

As a case study, we apply this method to develop and calibrate new items for a silent **Sentence Reading Efficiency** (SRE) task (Burkhardt et al., 2023), which we elaborate on in Section 2. We then propose an optimal-transport-inspired technique for generating parallel test forms with the new items by referring to a well-calibrated human-written test form. In doing so, we make the following contributions:

1. We address a novel task of automated item calibration for sentence reading efficiency, requiring a model capable of predicting both responses and response times.
2. We propose fine-tuning LLMs to estimate the properties of unseen items by simulating how past participants would have responded.
3. We demonstrate the effectiveness of marginalizing response-conditioned LLM predictions over a distribution of past participants to estimate their responses to new questions.
4. By automatically creating parallel test forms, we deploy our system to K-12 education and demonstrate its high quality through large-scale ($n = 234$) student evaluation.

Overall, our work advances the measurement of reading efficiency by introducing scalable methods to generate test items. We address the novel task of parallel test generation without collected responses

and approach parallel test construction with simulated responses as relaxed optimal transport.

```
Children can be sad.
True (Response time: slow)
You sleep on a log.
False (Response time: slow)
[...]
Sweaters can be made of coal.
False (Response time: very slow)
You can feed a hamster.
False (Response time: very slow)
You can fill a balloon with air.
```

Figure 3: Example prompt used for training and simulation. Given a prompt similar to this, the item-response simulator predicts the relevant item parameters – in our case, the student response and their log-response time.

## 2 Silent Sentence Reading Efficiency

How can we measure students' reading abilities? Oral Reading Fluency (ORF) is a widely used measure in research and practice, measuring words read correctly per minute (Fuchs et al., 2001; Domingue et al., 2022). Recently, researchers have examined the relationship between oral and silent reading fluency (Kim et al., 2011) and shifted focus to silent reading, as it is the most common form of reading for K-12 students. However, given the lack of observable verbal responses, it is a more challenging measurement task. Our task, silent Sentence Reading Efficiency (SRE), is an online, self-administered measure assessing the speed with which a student can read simple English sentences (Burkhardt et al., 2023). Modeled after other standardized silent reading fluency measures such as Test of Silent Reading Efficiency and Comprehension (TOSREC) (Johnson et al., 2011; Kim et al., 2012), SRE requires students to read sentences and respond whether each is True or False, which we

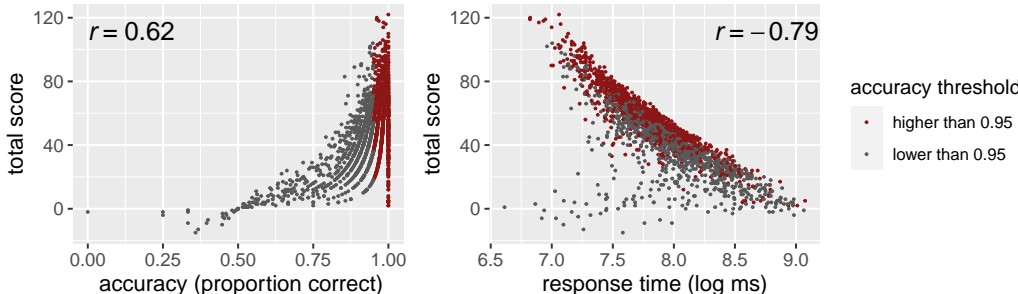

Figure 4: **Factors affecting reading efficiency scores**. (left) the relationship between each participant's total score and their response accuracy, (right) and their median response time in log scale. This indicates that response time is a more important factor that contributes to reading efficiency than accuracy. Note that the response times of participants with similar accuracy (e.g., >0.95) vary substantially and predict total scores.

refer to as the **item truth value** (see Fig. 3). A student responds to as many sentences as they can within three minutes, and the final score is the number of correctly answered sentences minus incorrectly answered ones. Unlike TOSREC, SRE targets reading efficiency with less emphasis on reading comprehension. This complicates item development, requiring syntactically and semantically simple sentences with only vocabulary that all school-age students should be able to read and understand.

We focus on SRE for three reasons. First, from an educational perspective, there is a high demand from schools to closely monitor students' reading development. Thus, it is important to develop diverse parallel test forms with identical difficulty and reliability. Manually authoring thousands of items and collecting the data to match test forms is extremely time-consuming and resource-intensive. Second, from a psychological measurement standpoint, SRE is a task where both accuracy and response time are crucial in determining ability. This characteristic does not align well with classic psychometric models, such as Item Response Theory (Lord, 1980), which focus on accuracy. Third, of particular relevance to the NLP community, measuring sentence-level reading fluency rather than comprehension poses a novel challenge, as traditional readability metrics and linguistic features fail to predict fluency accurately.

## 3 Training Dataset

We collect student data from 1st grade to high school by working closely with more than 30 diverse school partners in the United States for two school years (See Fig. 5 for breakdown by grade level). All data is collected under IRB guidelines. As part of the SRE validation study (Burkhardt et al., 2023), each student completed two 3-minute

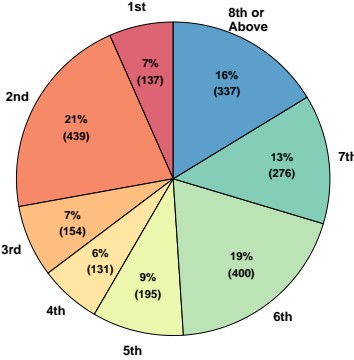

Figure 5: **Dataset grade distribution.** The grade distribution of students in the dataset discussed in Section 3, based on the grades they reported through the SRE app.

blocks with different test forms: one with TOSREC grade-specific sentences (maximum number of items varies by grade, but less than 70), and the other the *Lab test form*, consisting of 130 sentences curated by two human experts through an iterative design process. After filtering students with response times indicative of random guessing (median response times under 500 ms; $n = 108$), the final fine-tuning dataset includes 1962 participants with 183,782 responses. Fig. 4 indicates that the total score (i.e., the reading efficiency) of each student is more correlated with their median response time than how accurately they respond to each question, which underscores the importance of incorporating response time modeling to capture item difficulty during the item calibration process.

## 4 Methods

### 4.1 Item Generation

We use GPT-4 (OpenAI, 2023) to generate diverse sentences through zero-shot prompting and avoided giving specific examples. We first generate universally true sentences (temperature = 1, six completions, with at most 5000 tokens generated) and then transform each true sentence into a corresponding

false sentence by asking the model to change 1 or 2 verbs, nouns, or adjectives. After several rounds of iteration, we developed the following prompt — each rule after the third was added iteratively in response to observed failure cases:

```
Generate 150 sentences.
Rules:
1) The sentences have different length
   between 3 to 15 words.
2) 1st grade students should be able to
   read and understand the sentences.
3) Use only Kindergarten and high
   frequency vocabulary words to
   generate the sentences.
4) Make sure that each sentence is a
   sentence stating a universal fact
   that is immediately, obviously true.
5) If humans are used as the subject in
   a sentence, make sure it states a
   life experience that is true for
   everyone.
6) The sentence should not require any
   inferential comprehension skills to
   read nor understand.
7) The sentences should not be
   subjective nor describe opinions.
8) The sentences should not be centric
   to any country nor culture.
9) The sentences should be very diverse.
```

Excluding unexpected generation output (e.g., blank strings) and exact duplicates, this prompt gave us 761 true sentences. However, we found it generated few long sentences even though "3 to 15 words" was specified, so we prompted the model to generate an additional 100 sentences with an additional rule specifying "*at least 10 words in each sentence.*" We then used a prompt to transform each true sentence into a corresponding false sentence: *Transform each of the following sentences into false sentences by changing 1 or 2 verbs, nouns, or adjectives, and the sentences should be universally false.* Ultimately, this produced a corpus with 861 true and 861 false sentences.

### 4.2 Item Evaluation

For training, we fine-tuned an LLM to predict student responses conditioned on previous responses, which we refer to as an **item-response simulator**. We train on a manually authored, anonymized corpus of 1,962 participants collected by the lab, ranging from 1st grade to adulthood, containing response times and responses (i.e., true or false), described in Section 3. Specifically, each training example consists of a randomly-selected subset of a sampled participant's responses, which are arranged in random order. The model was then trained to predict the response and response time

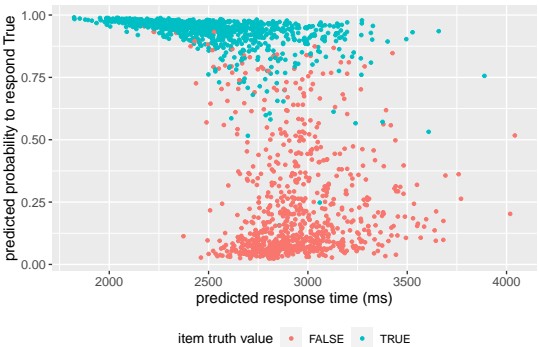

Figure 6: **Generated item simulated parameters.** We visualize the probability that the simulator assigns to a "true" student response for the GPT-4 generated sentences, colored by whether they came from the "true" or "false" set. For many "false" sentences scored as true, GPT-4 generated a sentence with the wrong truth value.

of the final item, conditioned on the previous items, as visualized in Figure 3. In our final model configuration, we use Low-Rank Adaptation (LoRA) (Hu et al., 2022) on a 13-billion parameter LLaMA model (Touvron et al., 2023) with 8-bit weights (Dettmers et al., 2022a,b) and a substantial 0.5 dropout on the adapter layers and final layer to mitigate overfitting. We include more detail on hyperparameters considered in Appendix E and discuss all differences between the hyperparameters used for the crowdworker and school student evaluations.

To apply this model for evaluating new items, we sampled previous participants and insert the generated item as the final item in a randomly sampled subset of their responses. Then, aggregating over many sampled participants per item, we calculate a mean and standard deviation response time for each sentence, as well as the expected proportion of simulated responses that were true or false for each sentence. Figure 6 visualizes the probabilities and response times simulated by the item-response simulator for the generated items, aggregated over a hundred sets of sampled student responses. Crucially, modeling the probabilities allows the simulator to identify ambiguous items – items for which a nontrivial percent of responses are expected to be incorrect. In Figure 6, false items with higher probabilities and true items with lower probabilities are more ambiguous. We include a qualitative analysis in Appendix A.2.

### 4.3 Parallel Test Form Construction

We first simulate student responses to both the lab test form and the generated items and try to identify a set of generated items that corresponded well to the lab test form. However, it is important

to be able to generate multiple distinct test sets – schools require multiple distinct parallel test forms to be administered throughout the year. A naive strategy would greedily pair the most similar items and then remove them when selecting new items for an additional test. But, this would ensure that successive tests would be gradually less similar to the original test and cannot ensure diverse items in each test. Instead, we duplicate the lab test form once per desired parallel test and find the best pairing between the lab and generated items. This is an unbalanced, constrained optimal transport problem, which we solve as follows: we first assign each duplicated lab item a probability of corresponding to a generated item, with no probability of true items being paired to false items and a term subtracted from the logits proportional to the distance between the lab and generated items.

We then minimize 1) the expected distances between the lab items and their paired generated items in the selected parameters (i.e., simulated response time), 2) the semantic similarity (over a threshold) within each copy of the dataset, and 3) the probability that the same generated item would be selected multiple times. This is a non-convex optimization problem, but by initializing the logits to reasonable initial values (e.g., logits proportional to distance), we found that the resulting simulated optimized tests converged and corresponded closely to the parameters that our model simulated for the lab-generated items. Figure 11 visualizes these simulated scores across two simultaneously generated, distinct test sets, as well as the ambiguous set of items for reference when optimizing for both accuracy and response time. Precisely, we sum over:

$$\ell_{\text{distance}} = \sum_{a=1}^{d} \sum_{i=1}^{n} \sum_{j=1}^{m} P_{aij} D_{aij}, \quad (1)$$

$$\ell_{\text{reuse}} = \sum_{a=1}^{d} \sum_{b=1}^{d} \sum_{i=1}^{m} \sum_{j=1, j \neq i}^{m} P_{a \cdot i} \cdot P_{b \cdot j}, \quad (2)$$

$$\ell_{\text{cosine}} = \sum_{a=1}^{d} \sum_{i=1}^{n} \sum_{j=1, i \neq j}^{n} P_{ai} C_{ij} P_{aj}, \quad (3)$$

for $P \in R^{d \times n \times m}, D \in R^{n \times m}, C \in R^{n \times n}$ where $n$ is the number of generated items and $m$ is the number of lab test form items, and $d$ is the number of test copies to optimize. $P$ is the probability that a lab test form item will be mapped to a given generated item, $D$ is the pairwise distance between the lab and generated item parameters, and $C$ is the semantic similarity between generated items.

Note that we only apply the optimal transport algorithm in crowdworker experiments, as the aim of the school student evaluation is primarily to demonstrate the usefulness of the item-response simulator in a collaborative setting, with more detail on the human-in-the-loop filtering included in Section 5.3. However, in future school deployments, we will use this algorithm, allowing us to jointly handle multiple optimization criteria. Because of the human-in-the-loop filtering setup, it was necessary to instead perform deduplication separately for the school student evaluation, discussed in Appendix C. We discuss implementation details further in Appendix E, and the full pipeline in Figure 2.

## 4.4 Additional Filtering

For the crowd-worker experiments, we filter the dataset for safety using GPT-4, discussed in Appendix D. For the school experiment, we manually review the questions for safety and ambiguity out of an abundance of caution and responsibility to provide high-quality items, discussed in Section 5.3.

## 5 Evaluations

### 5.1 Model Evaluation

**Experimental design.** For our computational model evaluation, we primarily validated our model by evaluating its ability to generalize to a random subset of 10% of the items in the dataset described in Section 3, not used for training. As also done for simulation, for each training example, we concatenated a subset of one student's responses to items they responded to, arranged in random order, visualized in Figure 3. We exclude the student response for the last item and fine-tune the model to predict it. As a detail, we also found that binning the input text corresponding to the student response times as "very fast", "fast", "medium", "slow", or "very slow" based on their quantile of overall data reduced overfitting. We believe this may reduce the model's ability to correlate specific sets of millisecond-level response times with student responses. Note that we calculated the bins based on the overall response time distribution because the variance across questions in the training dataset was much smaller than the variance across students. We include further item-level analysis for both the generated items and the item-response simulator's evaluations of these items in Appendix A.

**Results.** On our evaluation dataset, we find our simulator's item-aggregated predicted response

times are well-correlated with the item-aggregated true response times, with a correlation coefficient ($r$) of 0.50, and correspondingly, an $r^2$ of 0.25, and for predicted vs. actual aggregate probabilities, we found $r = 0.76$ ($r^2 = 0.58$) with a 25.4% RMSE.

## 5.2 Crowdworker Evaluation

**Experimental design.** Since school-aged students must be provided with carefully vetted tests with human supervision, we aim to utilize crowdworker evaluation to address questions that cannot be reasonably obtained through student evaluation alone:

1. Can the parallel AI test form (without any human filtering) reliably produce identical total scores compared with the well-calibrated, human-generated test form?
2. How well can the item-response simulator identify ambiguous generated items, and do these items actually challenge the reliability of the SRE measure?

To answer these questions, we develop a new version of the SRE task with three 3-minute blocks randomly ordered: human-generated sentences (Lab form), GPT-generated items filtered by the item-response simulator (parallel AI form), and GPT-generated items with ambiguous items identified by the item-response simulator (ambiguous AI form). To create the two AI-generated splits, we first divide the generated items according to the median accuracy of items in the training data and then follow our construction method in Section 4.3 on the unambiguous items and sample randomly for the ambiguous items. We recruited 50 participants via Prolific and paid ≈$15.00 per hour. 6 participants' results were found either incomplete or random guessing and were removed from the data analysis.

**Results.** The total score of each participant is counted by the total correct responses subtracted from the total incorrect responses. Figure 7 (top) shows that the total scores produced by the parallel AI test form achieve a high correlation ($r = 0.92$) with the scores produced by the Lab form. In addition, the overall difficulty of the parallel AI form is highly identical to the difficulty of the Lab form. Figure 7 (bottom) suggests that the test form with ambiguous AI items identified by the item-response simulator correlates much less ($r = 0.75$) than the parallel test form above. These comparisons suggest the item-response simulator, without human intervention, is able to flag unseen, ambiguous items that actually challenge test reliability.

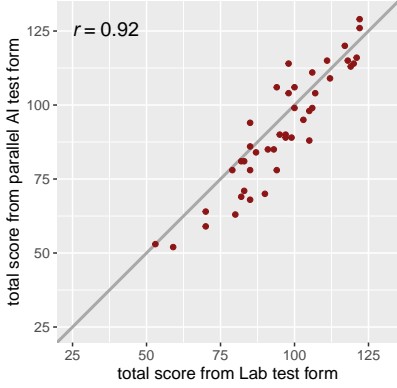

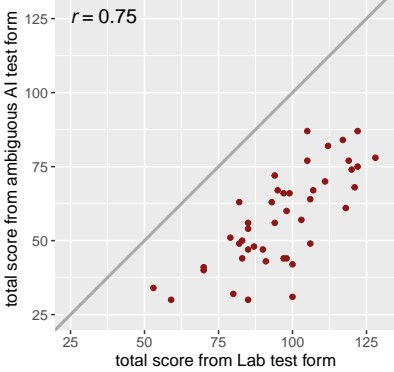

Figure 7: **Crowdworker scores.** Comparison of scores of Prolific participants on Lab test form with parallel AI test form (top) vs. ambiguous AI test form (bottom).

## 5.3 School Student Evaluation

**Experimental design.** The school student evaluation aims to answer three questions:

1. Can filtered GPT-generated items reliably produce identical total scores compared with human-generated items?
2. Can the item-response simulator outperform traditional readability metrics in predicting the response time and difficulty of unseen items?
3. And qualitatively, what are some items that the item-response simulator could predict well and what are some that it couldn't?

We use the item-response simulator to select the optimal 130 GPT-generated true sentences and 130 false sentences, 260 items in total. Then, to ensure the appropriateness of the test items for school-aged children to read and test, the authors, who are familiar with K-12 reading assessments, review and remove items that were still ambiguous (20 items, e.g., "A hill is flat and square."), could have an inappropriate interpretation (4 items), dangerous (2 items: "Babies drink gasoline." or "Soap is for eating."), required too much world knowledge (4 items, e.g., "A brick is made from clay") or are subjective (1 item, "A doll can be fun to play with."). Note that, in this school experiment, for robustness and out of concern for automation bias, we do not

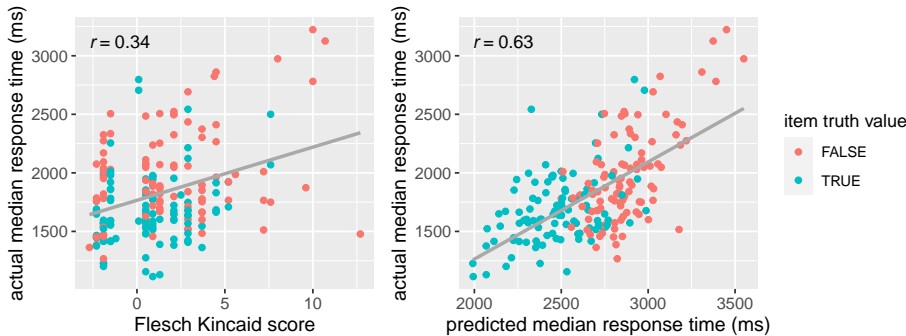

Figure 8: **Response time predictions**. Relationship between Flesch Kincaid grade vs. median response time (ms, left) and model predicted response times vs. actual median response times (ms, right) on the school student study.

Table 1: Parameters for human-generated items (Lab test form) and GPT-generated items with LLM and human filtering (AI test form). Accuracy and response times are based on item-response simulator simulations.

|     | Truth | Mean Len (Words) | Acc | Median RT (ms) | Std. RT (ms) |
|-----|-------|------------------|-----|----------------|--------------|
| Lab | False | 6.13 | 0.860 | 2926 | 202 |
|     | True  | 6.75 | 0.933 | 2691 | 392 |
| AI  | False | 5.02 | 0.872 | 2879 | 198 |
|     | True  | 5.42 | 0.947 | 2477 | 223 |

automatically filter the potentially offensive sentences as we do in the crowdworker experiments.

We then implement and deploy a new version of the SRE task with 2 blocks: human-generated sentences (Lab test form) and GPT-generated sentences filtered by the item-response simulator according to accuracy and response time (details in App. E.3) and then human experts (AI test form). 234 students in grades 2-8 from a California public school participated in this study, and three students were removed from the analysis due to a median response time under 500ms (i.e., random guessing).

In this version, students first see the human-generated block with a maximum of 130 items. Then, in the second block, students see 130 randomly-ordered GPT-generated sentences (from an item pool of 200 filtered items). The overall predicted item parameters based on the item-response simulator are shown in Table 1.

**Results.** There is a high correlation ($r = 0.93$) in terms of total scores between the filtered AI test form and the Lab form (Fig. 1). We also found that the two test forms match closely in difficulty, although we did not explicitly aim to match the identity line when creating the AI test form for schools.

In terms of predicting the unseen item parameters (accuracy and median response time), the item-response simulator outperforms the Flesch Kincaid method (Fig. 8) and accomplishes a task that the traditional psychometric models (e.g., Item Response Theory) cannot due to the unseen items (Table 2) - however, we note that prior work has attempted to model item difficulties using linear

Table 2: Model comparisons: prediction correlations

|                          | Accuracy                | Median RT (ms)        |
|--------------------------|-------------------------|-----------------------|
| Item Response Theory     | NA                      | NA                    |
| Flesch Kincaid           | -0.001 (-0.109, 0.106)  | 0.254 (0.149, 0.351)  |
| Item-Response Simulator  | 0.316 (0.215, 0.410)    | 0.509 (0.427, 0.586)  |

logistic test model (Sonnleitner, 2008; Stenner, 2022) and estimate IRT parameters from text data in other contexts (Ehara, 2018; Settles et al., 2020).

Fig. 9 showcases example items that are accurately and inaccurately predicted by the item-response simulator. Overall, the simulator can effectively screen difficult or ambiguous items.

## 6 Related Work

### 6.1 Item Generation

Earlier methods for the automatic item/question generation include rule-based or template-based approaches (e.g., Mitkov and Ha (2003); Flor and Riordan (2018)) and attempts to adaptively generate questions (Du et al., 2017). Additional work has focused on contexts in which generating high-quality questions (e.g., math word problems) requires step-by-step reasoning (Keller, 2021). In particular, question generation for passage-based reading comprehension (Agarwal et al., 2011; Stasaski et al., 2021; Heck and Meurers, 2022; Rathod et al., 2022; Zou et al., 2022) and for teaching second languages (Chinkina and Meurers, 2017; Srivastava and Goodman, 2021) have been especially well-explored.

More recently, White et al. (2022) demonstrate the effectiveness of using GPT-3 to create test items comparable to gold standard TOSREC test items for assessing reading fluency at first and eighth-grade levels. However, their approach still requires extensive human intervention and human evaluation for real-world use. Specifically, they demonstrate the feasibility of GPT-3 for item generation but do not develop an approach to calibrate items in terms of difficulty or filter items that would function poorly in the real world.

Difficult for students but easy for the simulator.

```
False: Music makes people sleep.
True: People wear sunglasses in
    sunlight.
True: Bananas are yellow when ripe.
```

Easy for students but difficult for the simulator.

```
False: Octopuses have two arms.
True: We use our stomach to digest food.
False: We wear belts to loosen pants.
```

Easy for both students and the simulator.

```
True: A turtle has a shell.
False: Books have pages with salsa.
True: Children play in the playground.
```

Difficult for both students and the simulator.

```
False: Stars are invisible at night.
False: Chairs have no legs.
True: Mushrooms grow in damp areas.
```

Figure 9: Example item difficulties as perceived by students and by the item-response simulator.

Relatedly, Srivastava and Goodman (2021) fine-tune a model to generate questions adaptively for teaching reverse translation by predicting whether a student will correctly complete a translation. While their study is a seminal prior work, it is not directly applicable to parallel test generation. First, we require a fixed item bank that can be closely reviewed before deployment because our items will be given to children and because we cannot have extra latency. Moreover, the best models for many zero-shot item generation tasks cannot be fine-tuned by the public, but prior work suggests that small models can effectively validate much more powerful models (Cobbe et al., 2021). Finally, we focus on providing reliable assessment, while their approach is only applicable to instruction.

## 6.2 Item Evaluation

Although item generation has broadly received more attention from the NLP community than item evaluation, there are several influential works in this area, especially leveraging language models. Language models have been used as a proxy for reading, with prior work highlighting the limited correlation between human and language model readability: studies like Schwarm and Ostendorf (2005) and Si and Callan (2001) demonstrate that the perplexity of a language model is a powerful indicator of the likelihood that a sentence is generated from a corpus of a given difficulty. These works are also related to knowledge tracing, i.e.,

modeling student knowledge (Piech et al., 2015; Abdelrahman et al., 2023). More recent studies, such as Benzahra and Yvon (2019) and Martinc et al. (2021), contrast this finding, noting that language model perplexity and other statistical metrics were imperfect estimators of readability, but metrics derived from them performed well across datasets. Importantly, predicting reading fluency, while naturally related to reading comprehension, is a distinct and under-explored area.

## 6.3 Parallel Test Form Construction

The body of work on constructing parallel tests spans decades (Armstrong et al., 1992, 1994; Gibson and Weiner, 1998; Sun et al., 2008; Ignjatović et al., 2021). However, as most of these works point out, identifying the optimal pairing is an NP-hard problem. To circumvent this, the works often rely on item selection heuristics. For example, Armstrong et al. (1992, 1994) calculate the degree to which each item is correlated with the total test score and then attempt to generate tests with similar expected distributions of final scores. Similarly, Sun et al. (2008) uses a genetic algorithm to maximize the similarity in the expected information gain of each item across a set of tests. In contrast, we pose this as a relaxed optimal transport problem – we frame our optimization as a differentiable, probabilistic relaxation of this underlying optimal transport problem and solve it directly. This allows us to incorporate other optimization criteria that are important but rarely seen in parallel test literature, such as item diversity and truth parity constraints.

## 7 Conclusion

Our study presents a new framework to generate test items and select them according to their effectiveness to assess students by leveraging large language models and previous student responses. We use silent sentence reading efficiency assessment, a task with implications for the millions of students who take such tests annually, to illustrate the utility of this framework by generating new test items, predicting the difficulty and ambiguity of unseen items, and creating multiple parallel test forms that have comparable quality and appropriateness to human-generated test forms as well as remain appropriately difficult for students. We ultimately validate our approach with a wide range of evaluations, including more traditional machine learning metrics, crowdworker evaluations, as well as a real-world deployment in K-12 classrooms.

## Limitations

While our case study demonstrates the effectiveness of using large language models and previous student responses, it carries several limitations:

**Generalization to other types of tests and questions:** The framework presented in this case study focuses primarily on assessing students' reading efficiency. However, we have not yet demonstrated that our approach can easily generalize to other kinds of tests and assessments. No aspect of this approach is inherently dependent on the SRE task, but more research is needed to investigate the applicability of our methods to a wider range of educational tests. Although multiple-choice questions are not typically used in real-world silent sentence reading fluency evaluation (Bloomquist, 2017), we believe with a comparable amount of data, we could replicate this success for other test formats such as multiple-choice questions.

**Filtering and evaluation in school student evaluation:** Due to ethical considerations, certain aspects of our study, such as automated filtering of ambiguous or inappropriate items, could not be deployed in the school student evaluation. Consequently, the items in the school student evaluation had to be carefully reviewed by experts before administration. This highlights the need for more robust and reliable methods for filtering and evaluating generated items in real-world educational settings. Although we now have validation supporting the item-response simulator, deploying aspects like automatic filtering will require further study, not only to assess accuracy and sensitivity but also to mitigate automation bias and risks.

**Fine-tuning and data limitations:** The item-response simulator was fine-tuned on data collected from diverse schools in the United States, but the model is trained on a uniform distribution of these students. However, the students whose responses are used for training and simulation may differ demographically from the schools to which the tests are deployed. Thus, the model's performance in predicting item difficulty and ambiguity may not generalize to all populations or school contexts. Moreover, we have not quantified the degree to which additional generated examples from GPT-4 continue to be novel – language models fine-tuned using reinforcement learning from human feedback (RLHF) are believed to suffer from mode collapse (Zhu et al., 2023), so ensuring that generated items continue to be meaningfully different is essential.

**Reliance on closed and restricted LLMs:** Our study uses GPT-4 for generating and filtering test items. However, access to GPT-4 may be expensive if generating many thousands of items. In addition, we fine-tuned LLaMA (Touvron et al., 2023), but LLaMA's license does not support commercial use. As a result, the exact approach in this paper cannot be applied in commercial contexts. Fortunately, LLaMA replications are underway, and artifacts have already been released (Geng and Liu, 2023).

## Ethics Statement

There are naturally many ethical considerations in the context of this work. First, all handling of student data must be extremely careful and considerate of the privacy of the students. In this work, we have taken care to ensure that the data used is not personally identifiable and that the data is used appropriately. We have also acted in accordance with the guidelines of our IRB. At all points, before presenting data to humans, especially to children, multiple rounds of review and analysis were performed to ensure that the data was appropriate. The value of our work is to largely reduce the burden of the test creation and invite humans to act as the final safeguards and experts to be responsible for reviewing as few items as possible.

As for the ethical implications of the work itself, there are several to consider. First, language models exhibit and potentially exacerbate biases that are already present in society. Humans, of course, also exhibit these biases, but by automating this pipeline, we may reduce the possibility of human intervention to correct for these biases. Second, the ability to use language models in this context may result in an emphasis on tests being developed that are compatible with language models – however, aspects like visual and phonetic information are valuable for reading fluency evaluation and many other tasks, and we should be mindful to avoid tailoring tests closely to language models' strengths.

Finally, while the ability to efficiently generate test items could lower the barrier to universal assessment and lead to more equitable assessment policies, it's important to proceed with caution and keep humans in the loop at each stage – particularly when it pertains to educational assessment in young children. Thus, as we begin implementing AI-based assessment systems at scale, we advocate for proceeding with caution, keeping educational experts in the loop at each stage, and keeping an eye toward equity as we strive for efficiency.

## Acknowledgements

We would like to thank the schools that partnered with us on this research. We thank the Stanford NLP community for their helpful feedback on the abstract of this paper. We appreciate Benjamin W. Domingue and the reviewers for their helpful feedback on the manuscript and Sang T. Truong for highlighting related work. This work was funded by grants from the Stanford-Sequoia research collaborative, Stanford Neuroscience:Translate program, Microsoft, and the Eunice Kennedy Shriver National Institute of Child Health and Human Development R01HD095861 to J.D.Y, and NSF IIS-2247357 to D.Y.

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

## A Model Analysis

### A.1 Analyzing GPT-4-generated Sentences

GPT-4 and, more broadly, language models fine-tuned based on human preferences have well-known challenges with mode collapse – although higher temperatures theoretically encourage diversity, this training step causes models to be more likely to express a single "modal" opinion (Santurkar et al., 2023). We observed a similar pattern, with a much higher degree of similarity across different instances of model generations as opposed to across items within a single generation. This was part of the motivation for encouraging the model to generate many diverse sentences in a single prompt instead of performing many independent calls. The following were some of the highly similar sentences that remained even after the item-response-simulator-based filtering, as determined by the sentence embedding model we used to filter the sentences further:

```
A stove is for cooking food.
A stove is for cooking.

Fruit grows on trees and plants.
Fruits grow on trees and plants.

Cows give us milk to drink.
Cows give us milk.

A toothbrush cleans our teeth.
Toothbrushes clean our teeth.

Apples grow on apple trees.
Apples grow on trees.

A bicycle has two wheels.
A bike has two wheels.

Vegetables are good for our health.
Vegetables are healthy food.

Kites can fly in the sky.
Kites fly in the wind.

A boat floats on water.
A boat goes on water.

Ice cream is cold and sweet.
Ice cream is cold.

Toothpaste cleans our teeth.
Toothpaste helps to clean teeth.
```

### A.2 Analyzing item-response-simulator Predictions

We observe several interesting patterns in our item-response simulator, especially in combination with GPT-4. For example, as mentioned in Figure 6, many of the truest "false" sentence sentences generated by GPT-4 were, in reality, true and many of the least false or most uncertain "true" sentences were in fact somewhat ambiguous. Note this trend was particularly pronounced for the sup-posedly "false" sentences, where the item-response simulator determined that the following sentences were the least false (excluding repeated sentences) – most are at least arguably true:

```
[Keys lock doors. Ducks honk and fly.
    Boats sink in water. We bake food
    in ovens. A farmer eats food. Rain
    falls from the ground. Shadows form
    in darkness. We cut food with
    spoons. Umbrellas are used to keep
    us wet during rainstorms. Sailboats
    have a sail to avoid the wind and
    stay still on the water. Lightning
    comes after thunder. Water is
    frozen. Ants are large insects that
    can carry heavy loads and work
    together. We eat when we are full.
    People use tools to break things. A
    shirt uncovers our body. Buses take
    us places slowly. Families swim
    together. Autumn leaves stay on
    trees. Blankets help keep you cold.
    A bathtub releases water. A nap
    makes us tired.]
```

The mistake of making an explicitly, unambiguously incorrect truth judgment was not an observed failure case for generated "true" sentences: for these sentences, the model was unlikely to generate something outright false and more likely to generate something ambiguously true or only subjectively true. For example, the following true examples were the ones where the model was most uncertain - many of them contain statements that are not universally true, confusing, or difficult to assign a truth value. The following "true sentences" were among the ones where the model was least certain of the student response:

1. "*Foxes are orange and fast..*" Foxes come in a variety of colors and fast is subjective.
2. "*A hill is small and round..*" Relative to mountains, sure, but this is not universally true.
3. "*Clocks have twelve numbers.*" What about digital clocks and 24-hour clocks?
4. "*The moon changes shape..*" The moon appears to change shape in the sky, but it does not actually change shape.
5. "*Dolphins are not fish..*" While true (disregarding folk taxonomy), this is clearly a world-knowledge-heavy question.
6. "*Rice grows in water..*" Again, this requires an unreasonable amount of world knowledge.

## B Preliminary Crowdworker Study

We performed an initial crowdworker study where we performed our optimization to match each copy

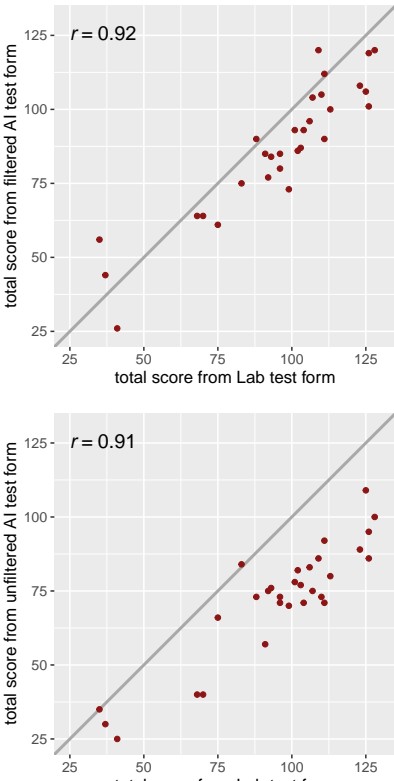

Figure 10: **Preliminary crowdworker study results.** Results from the preliminary crowdworker study. While the unfiltered items matched the difficulty more poorly than the filtered items, we incorporated these observations when constructing our main crowdworker study.

of the lab dataset to a generated item in terms of both accuracy and response time. We observed that the difficulty did not perfectly match that of the lab dataset, as represented by the identity line, although it corresponded much more closely than the unfiltered dataset. When comparing the ground truth data to the predictions, the cause of this was clear: the few lab items that the model identified as ambiguous were substantially less ambiguous than predicted. However, for the generated items, they were actually ambiguous. This transformed false negatives into true negatives in the dataset, harming performance. As a result, for our final prolific study and for the school evaluation, we filtered by the median accuracy but did not optimize it when matching generated and lab items.

Figure 10 shows both filtered and unfiltered test forms correlate well with the lab test form. However, there are important caveats: first, prolific participants represent a substantial distribution shift for the model which was trained primarily on K-12 students – they performed far better on average;

second, we found that the difficulty of the filtered corpus matched the lab corpus far more.

## C  Naive Deduplication

After training these models and validating that their predictions appeared reasonable, we filtered the items to include only ones where the predicted response time was closest to the trendline relating sentence length (in words) to response time. We further filtered them by predicted accuracy, selecting only the ones that the model was most confident students would answer correctly (in particular, we chose the top 200 true and false sentences). Finally, we used sentence embeddings generated for each sentence by the "paraphrase-mpnet-base-v2" model to select the most similar pairs of sentences (in terms of absolute cosine distance, to also capture sentences that were similar in meaning but opposite in truth value) (Reimers and Gurevych, 2019). If the sentences had the same truth value, we would remove one at random. If they were different, we would remove the one corresponding to the majority class. This ensured we were left with a roughly equal number of both true and false sentences, with 260 in total.

However, this carried a limitation: by deduplicating before selecting items, we may eliminate potentially good items. By deduplicating after selecting items, we must anticipate the proportion of items that must be deduplicated in advance. Motivated partly by this, we then explored techniques to simultaneously optimize semantic similarity alongside test set quality in a less heuristic way.

## D  Additional Filtering

To remove potentially inappropriate items for classroom settings, we further prompt GPT-4 to evaluate the appropriateness of their own generation (with temperature = 0.1, and at most 1000 tokens generated). Note that this was only used for the crowdworker experiments – for the school student evaluation, we performed this filtering manually out of an abundance of caution. We use the following prompt:

```
Return the following true or false
    sentences if it is potentially
    offensive or dangerous for children
    to read.
```

## E Implementation Details

### E.1 Item Evaluation

We explored various training methods, including Low-Rank Adaptation (LoRA) (Hu et al., 2022) on a model using 8-bit weights (Dettmers et al., 2022a,b), optimizing only a subset of the weights represented as an additive term (a matrix constructed as the product of two vectors that is added to the original linear layer weights). We also explored training a linear head on top of a pre-trained model - initially, we used the mean of the final hidden state embeddings, but later observed that this underperformed, as the considered models were autoregressive and the prediction is only possible from the final example, and not the participant-specific few-shot examples. Instead, we used the final example's final hidden state, which performed better. Each training example corresponds to a subset of a sampled participant's responses – however, when we simulate responses to an item for evaluation, we sample many participants, predict their responses to the new item, then aggregate the predictions.

For initial exploration, we primarily explored a variety of Open Pre-trained Transformer (OPT) language models with between 125 million and 6.7 billion parameters (Zhang et al., 2022). For the final implementation, we primarily considered LLaMA models, varying from 7 billion parameters to 65 billion parameters (Touvron et al., 2023). The largest model we fine-tuned with LoRA had 13 billion parameters and the largest model we used with a linear classifier had 65 billion parameters, leveraging four 40GB A40 GPUs. In practice, we found that these largest models were too slow for our purposes, ultimately using a 13-billion parameter LLaMA model.

We ultimately selected a constant learning rate of $2e^{-5}$ with a batch size of 32 sets of items, including a random sample of up to 30 student item-response pairs (fewer only if the student responded to fewer items). We use Low-Rank Adaptation (LoRA) (Hu et al., 2022) on a model using 4-bit weights (Dettmers et al., 2022a,b, 2023), optimizing only a subset of the weights. We trained our model for 600 steps, each of which contained 32 items, with gradient accumulation over a batch size of 4. Training the model required approximately 6 hours, while simulating all of the GPT-generated items with 100 previous participants required approximately 24 hours.

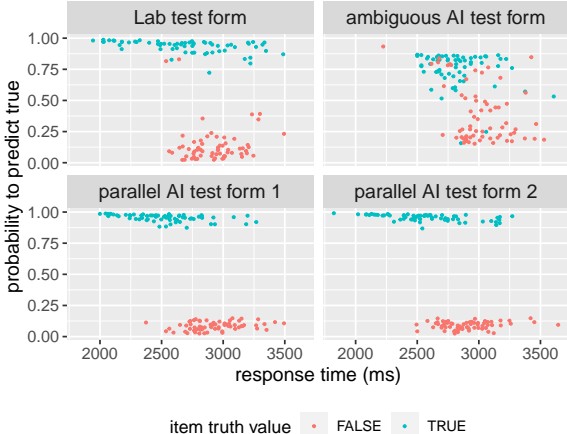

Figure 11: **Test form item distribution comparison.** Simulated response time and accuracy distributions for generated test forms.

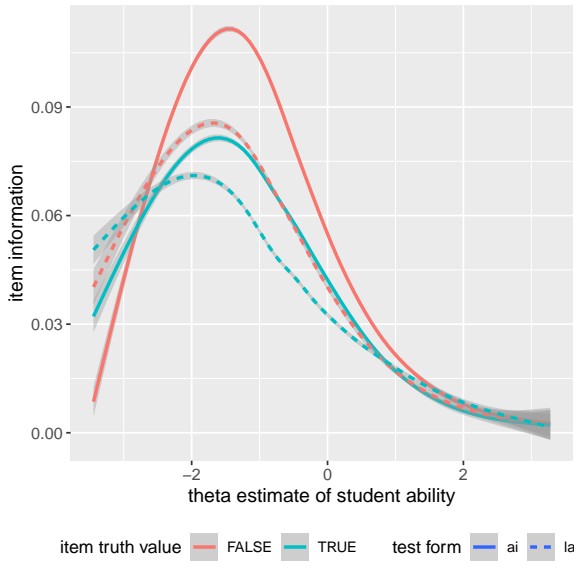

Figure 12: **Item information comparison.** Item information based on 2-parameter logistic IRT.

### E.2 Parallel Test Form Construction

We perform our optimization over each item-pair's logits with Adam with a learning rate of 0.1 and evaluate item similarity using the absolute cosine similarity between their sentence embeddings from SentenceTransformers's `paraphrase-mpnet-base-v2` (Reimers and Gurevych, 2019), disregarding items with absolute similarity below 0.5. Note that we initially attempted to use Feydy et al. (2019)'s differentiable optimal transport library, `geomloss`, but unbalanced problems require a reach hyperparameter to be specified, and we found that no reach works across all points. Figure 11 showcases the predicted response time and accuracy distributions in each test forms.

### E.3 School Student Evaluation Filtering

Based on feedback from experts, for our school student evaluation, we initially filtered using two threshold criteria: first, accuracy must be greater than 85%, which corresponds to the median item accuracy in our dataset; second, the response time should be within a standard deviation of the mean response time per word (disregarding the intercept – i.e., the extrapolated amount of time we expect a participant to take for an item with no length).

## F  Item Information Analysis

By fitting a 2-parameter logistic item response theory model to the student responses to both the Lab and AI-filtered test forms, we obtain the Fisher information (Fisher, 1925) for each item. Figure 12 shows that GPT-generated false sentences offer higher item information than human-generated false sentences, while there is no significant difference for true sentences. Although IRT is not the perfect psychometric model for the SRE task, this finding is encouraging to show the potential of the GPT-generated items.