# OpenReview forum: "Generating and Evaluating Tests for K-12 Students with Language Model Simulations: A Case Study on Sentence Reading Efficiency"
_EMNLP/2023/Conference — EMNLP 2023 Main_

### Official Review · Reviewer_CGAU · 2023-08-04

**Soundness:** 4

**Excitement:**

3: Ambivalent: It has merits (e.g., it reports state-of-the-art results, the idea is nice), but there are key weaknesses (e.g., it describes incremental work), and it can significantly benefit from another round of revision. However, I won't object to accepting it if my co-reviewers champion it.

**Paper Topic And Main Contributions:**

This paper introduced an educational test item generation method utilizing Large Language Models (LLMs). The proposed method involved the development of prompts stating several conditions. The primary contribution of this study lay in exploring the effective use of LLMs, and this study demonstrated their capacity to generate test items and evaluate the difficulty of test items. The research specifically targeted K-12 students' reading efficiency, focusing on testing the reliability of test items developed using the proposed method. The study compared the test results obtained from the proposed method with those generated manually. Additionally, the test-item difficulty was assessed by comparing with readability metrics. The findings indicated a strong correlation between the test results obtained through the proposed method and the manually produced tests.

**Reasons To Accept:**

This paper effectively demonstrated the benefits of LLM in producing silent reading efficiency test items and evaluating their difficulty. Particularly notable is the finding that the use of LLMs maintained the reliability and validity of manually produced tests, as shown in Figure 1, where the scores from AI test form and those of Lab test form had strong correlation. This sudy also explored the process of calculating the correspondence probabilities between lab items and AI-generated items.

**Reasons To Reject:**

The focus on True/False questions and response time measurement is well-suited for the reading efficiency test. However, it would be beneficial to consider incorporating a wider range of question types and options in an adaptive test format. This expansion would facilitate a more comprehensive evaluation and bring your research in line with standard educational assessment practices. Additionally, when delving into the process of calculating correspondence probabilities between lab items and AI-generated items, please provide a more detailed explanation of the training data's validity. In cases where the training data is incomplete or contains biases, there's a potential risk of obtaining biased results.

**Reproducibility:**

3: Could reproduce the results with some difficulty. The settings of parameters are underspecified or subjectively determined; the training/evaluation data are not widely available.

**Reviewer Confidence:**

3: Pretty sure, but there's a chance I missed something. Although I have a good feel for this area in general, I did not carefully check the paper's details, e.g., the math, experimental design, or novelty.

---

> ### Author Rebuttal · Authors · 2023-08-28
>
> Thank you for the valuable feedback! We appreciate your suggestions for improving the scope and agree that these would be impactful future directions. To clarify, the choice to use true/false sentences as test items was to closely match standard reading efficiency assessments, e.g., TOSREC, the Woodcock-Johnson Sentence Reading Fluency task, and our SRE. A core aspect of human-centered NLP is the need to be specific about who we are serving and how to efficiently incorporate human data to develop useful NLP applications. By collecting 2 years of responses from over a thousand students from diverse K-12 populations, we were able to fine-tune this model to scale the original human-written test. The specificity of our data and our evaluations helped ensure our contribution is efficient and fair for the populations we aim to serve. As the reading patterns of adult crowdworkers are naturally different from those of e.g. primary school students, we aim to use crowdsourcing only for confirmatory purposes where it would be impossible to evaluate with our served population.
>
> However, on the topic of other test formats:
> 1. Note that multiple-choice questions are not typically used in real-world silent sentence reading fluency evaluation, but are used in tests like reading comprehension evaluation [1,2].
> 2. However, we believe that with our method and a comparable amount of data, we could replicate this success for other test formats such as multiple-choice questions.
> 3. If we incorporate existing methods for automatic grading of open-ended responses [3], we expect we could even generate high-quality short-response items.
>
> We agree that exploring our method’s generalizability would be a valuable contribution. We will incorporate these points into the paper, and we’d be happy to elaborate on additional potential future directions. The suggested extensions would make for exciting follow-up works and deserve more than experiments we could only have run with crowdsourced participants.
>
> Finally, the primary reason we opted for a non-adaptive assessment for this study was to create parallel test forms that can produce scores consistent with our human-written gold standard test. In psychometrics, it is crucial to validate the reliability of alternative assessment (in our case, the AI test form) first, before scaling the test to a more adaptive format [4]. Additionally, using a predefined item bank rather than generating items on the fly allows us to (1) review all generated items to ensure their safety and developmental appropriateness for school-aged participants, and (2) to run the assessment entirely in-browser without additional latency or significant participant hardware requirements. But at the same time, we agree that an adaptive test would provide significant additional value.
>
> [1] “An examination of the relationship of oral reading fluency, silent reading fluency, reading comprehension, and the Colorado State reading assessment” Bloomquist, 2017
> [2] “Relationships of Three Components of Reading Fluency to Reading Comprehension” Klauda and Guthrie, 2008
> [3] “Exploring Automatic Short Answer Grading as a Tool to Assist in Human Rating” Condor 2020
> [4] “Overview of Test Assembly Methods in Multistage Testing” Zheng et al. 2014 (in Computerized multistage testing)

---

### Official Review · Reviewer_dxrP · 2023-08-05

**Soundness:** 4

**Excitement:**

4: Strong: This paper deepens the understanding of some phenomenon or lowers the barriers to an existing research direction.

**Paper Topic And Main Contributions:**

This study involves reading test generation and assessment for K-12 students.
LoRA and LLaMa were used for reading comprehension test generation. For the evaluation, because it is difficult to collect and evaluate children who are actually at the K-12 stage, the reading test is evaluated by crowdsourcing the reading test, manually evaluating it, and comparing the K-12 students' scores on the existing test to their crowdsourced scores.

**Reasons To Accept:**

They are approaching the training in the best-performing way they can think of at this time for a clear objective: a combination of LLaMa and LoRA. The paper also evaluates in the best way conceivable at this time, despite the fact that it is also difficult to evaluate.

**Reasons To Reject:**

In Table 2, it is true that Item Response Theory cannot be evaluated for unseen samples. However, there are several NLP studies that estimate IRT difficulty parameters from texts. In terms of studies on reading materials such as this study, the following studies would be relevant. It would be better to discuss the relevance of these previous studies.

Machine Learning-Driven Language Assessment (Settles et al., TACL 2020)
https://aclanthology.org/2020.tacl-1.17/

Building an English Vocabulary Knowledge Dataset of Japanese English-as-a-Second-Language Learners Using Crowdsourcing (Ehara, LREC 2018)
https://aclanthology.org/L18-1076/

**Reproducibility:**

4: Could mostly reproduce the results, but there may be some variation because of sample variance or minor variations in their interpretation of the protocol or method.

**Reviewer Confidence:**

4: Quite sure. I tried to check the important points carefully. It's unlikely, though conceivable, that I missed something that should affect my ratings.

---

> ### Author Rebuttal · Authors · 2023-08-28
>
> Thank you for your encouraging and supportive review, as well as the many insightful comments!
>
> First, we were hoping to highlight a key detail related to your summary. It is true that we used crowdsourced responses for part of the evaluation for this study. However, a primary contribution of our work that we would like to emphasize is that we indeed administered our test (with extensive review and careful oversight) to 234 students actually at the K-12 stage at a California public school, who took both our AI-generated test and the human-generated test. Figure 1 in the paper is based on this data. We realize that this may have been unclear and will adjust the abstract and Figure 1 accordingly, to clarify that these were actual K-12 students.
>
> In addition, we sincerely thank you for highlighting these relevant works - we will reference them and clarify in the paper that, while IRT itself cannot be evaluated for unseen items, prior work has aimed to estimate IRT parameters.
>
> It is true that the work from Settles et al. (2020) demonstrates that IRT $\theta$ parameters have a rank-correlation with their CEFR-derived predictions. However, their paper also indicates that their passage rank is well-predicted by a linear regression of relatively simple factors like words per sentence and characters per word, factors which are combined in the Flesch-Kincaid score. As indicated in our Table 2, Flesch-Kincaid was not predictive of our metrics. We also emphasize that a benefit of our approach is the ability to use a model fine-tuned on many students to simulate the responses of a subset of students. In addition, Ehara et al. (2018), while absolutely relevant and honestly quite prescient, focused on mapping individual word embeddings to IRT parameters and thus would be potentially difficult to apply to this sentence evaluation.

---

### Official Review · Reviewer_qWhv · 2023-08-05

**Soundness:** 4

**Excitement:**

3: Ambivalent: It has merits (e.g., it reports state-of-the-art results, the idea is nice), but there are key weaknesses (e.g., it describes incremental work), and it can significantly benefit from another round of revision. However, I won't object to accepting it if my co-reviewers champion it.

**Paper Topic And Main Contributions:**

The study focuses on silent sentence reading efficiency assessment and presents a framework for generating items, simulating a student’s responses, and producing parallel tests. The work is well-thought, well-organized, and presents comprehensive evaluations.


**Reasons To Accept:**

* The work is well-written and provides thorough details.
* The study presents comprehensive evaluations.

**Reasons To Reject:**

* The work lacks technical novelty.
* The weak relevance to EMNLP.

**Reproducibility:**

4: Could mostly reproduce the results, but there may be some variation because of sample variance or minor variations in their interpretation of the protocol or method.

**Reviewer Confidence:**

3: Pretty sure, but there's a chance I missed something. Although I have a good feel for this area in general, I did not carefully check the paper's details, e.g., the math, experimental design, or novelty.

---

> ### Author Rebuttal · Authors · 2023-08-28
>
> Thank you for the thoughtful review and feedback on our work! We appreciate your comments on the comprehensiveness of our evaluations and the detail provided.
>
> Regarding your concerns about novelty and relevance to EMNLP, we were hoping to highlight that this work was submitted to the new Human-Centered NLP track; while we readily acknowledge that our primary contribution is not the method’s technical novelty from a core NLP perspective (i.e., we’re not proposing a new model architecture), we’d like to emphasize that our key contribution is in showcasing a technique for a novel human-centered application, and in demonstrating its real-world applicability.
>
> With that said, we’d like to highlight a few key points of technical novelty:
> 1. Our key technique is marginalizing response-conditioned LLM predictions over a distribution of past participants. This can be applied anytime you want to know how people might respond to new, in-distribution questions.
> 2. Framing parallel test construction as optimal transport has applications anywhere high inter-test reliability is necessary.
> 3. Fine-tuning for this task is nontrivial, and many techniques should be transferable. We address questions like whether to jointly learn psychometric properties and how to prevent LLM overfitting while maintaining performance on limited human data.
>
> Moreover, we hope this paper can help bring together psychometrics and NLP, where simulating student responses has potential in pre-evaluating the quality, efficiency, and equity of the assessments/measures. Many NLP advances have come about this way: techniques to solve subfield-specific problems often find broader applications. Especially for human-centered NLP tasks where individuals may disagree on the correct answer or label, fine-tuning language models to simulate prior participants’ responses conditioned on their past responses has broad application. Not only can it help to model complex overall distributions of responses, but it can also provide insights into the causes of differences in responses. This could allow us to ask nuanced questions about the person-to-person sources of variation.
>
> We will emphasize these technical aspects and novelty in our revised version. Broadly, we are enthusiastic about the growing relevance of human-centered NLP to the wider NLP community.

---

### Meta-Review · Area_Chair_sTcL · 2023-09-19

**Recommendation:** 4

**Metareview:**

This paper leverages LLMs to significantly reduce the effort required for generating tests for K-12 students. Reviewers are highly positive about the approach and evaluation, giving high soundness scores (4). Concerns are raised regarding the lack of technical innovation (qWhv), missing related work (dxrP), and the limited range of the question types and options (CGAU).

I recommend accepting this paper to the main conference. Overall, this is solid research that illustrates a very reasonable use case for LLMs. The concerns are minor and do not overshadow the contributions. However, the authors should take them seriously and make an effort to address them in the next version.

---

### Decision · Program_Chairs · 2023-10-07

**Decision:**

Accept-Main

**Comment:**

This paper leverages LLMs to significantly reduce the effort required for generating tests for K-12 students. Reviewers are highly positive about the approach and evaluation, giving high soundness scores (4). Concerns are raised regarding the lack of technical innovation (qWhv), missing related work (dxrP), and the limited range of the question types and options (CGAU).

I recommend accepting this paper to the main conference. Overall, this is solid research that illustrates a very reasonable use case for LLMs. The concerns are minor and do not overshadow the contributions. However, the authors should take them seriously and make an effort to address them in the next version.